# Chirality-driven orbital magnetic moments as a new probe for topological magnetic structures

Manuel dos Santos Dias[1], Juba Bouaziz[1], Mohammed Bouhassoune[1], Stefan Blügel[1] & Samir Lounis[1]

When electrons are driven through unconventional magnetic structures, such as skyrmions, they experience emergent electromagnetic fields that originate several Hall effects. Independently, ground-state emergent magnetic fields can also lead to orbital magnetism, even without the spin–orbit interaction. The close parallel between the geometric theories of the Hall effects and of the orbital magnetization raises the question: does a skyrmion display topological orbital magnetism? Here we first address the smallest systems with nonvanishing emergent magnetic field, trimers, characterizing the orbital magnetic properties from first-principles. Armed with this understanding, we study the orbital magnetism of skyrmions and demonstrate that the contribution driven by the emergent magnetic field is topological. This means that the topological contribution to the orbital moment does not change under continuous deformations of the magnetic structure. Furthermore, we use it to propose a new experimental protocol for the identification of topological magnetic structures, by soft X-ray spectroscopy.

[1] Peter Grünberg Institut and Institute for Advanced Simulation, Forschungszentrum Jülich & JARA, D-52425 Jülich, Germany. Correspondence and requests for materials should be addressed to M.d.S.D. (email: m.dos.santos.dias@fz-juelich.de) or to S.L. (email: s.lounis@fz-juelich.de).

The magnetic moment has two contributions: the spin magnetic moment and the orbital magnetic moment, which are due to lifting of spin and orbital degeneracy, respectively. The familiar mechanism lifting the orbital degeneracy is the spin–orbit coupling (SOC), $\mathcal{H}_{SOC} = \xi \mathbf{L} \cdot \mathbf{S}$, which leads to an orbital moment ($\mathbf{L}$) tied to the spin moment ($\mathbf{S}$). $\xi$ is the strength of the SOC. A more general picture emerges by analogy with the classic orbital moment, a closed current loop. In quantum physics, ground states hosting bound currents also require lifting of orbital degeneracy. Such ground-state currents have been proposed for magnetic structures where the magnetic moments do not all lie in the same plane, that is, with nonvanishing scalar spin chirality, $C_{123} = \mathbf{S}_1 \cdot (\mathbf{S}_2 \times \mathbf{S}_3) \neq 0$ (refs 1–3). Ground-state magnetic structures with $C_{123} \neq 0$ can be stabilized by SOC-driven interactions, such as the magnetic anisotropy or the Dzyaloshinskii–Moriya interaction, or by interactions independent of SOC, such as frustrated bilinear exchange interactions[4] or higher-order exchange interactions, exemplified by the biquadratic and four-spin interactions[5]. Remarkably, electronic structure calculations have predicted orbital moments without SOC[6–8], but the properties of the orbital moments and their usefulness are unexplained and unexplored.

Electrons flowing through a magnetic system couple to emergent electromagnetic fields, leading to several Hall effects[9–12]. Noncoplanar magnetic structures ($C_{123} \neq 0$) have a finite emergent magnetic field in their ground state[10], with the most well-known examples being skyrmions[13–16]. The scalar spin chirality, $C_{123}$, is closely related to the emergent magnetic field[11,12], a natural concept in the geometric theories of the Hall effects and orbital magnetization[17]. The net flux of the emergent magnetic field in a skyrmion is quantized and corresponds to the topological charge or skyrmion number of the magnetic structure[10]. Ref. 7 suggested a link between the emergent magnetic field and orbital moments, but did not atttempt to investigate it. Possible consequences of the topological properties of the magnetic structure on the orbital magnetism remain open.

Establishing the topological character of a given magnetic structure experimentally is challenging. The topological Hall effect, driven by the emergent magnetic field, is the most direct signature, but all other contributions to the Hall signal have to be unpicked and carefully subtracted[9,18,19]. The topology can also be ascertained via full knowledge of the three-dimensional magnetic structure, which can be mapped via small-angle neutron scattering[14], Lorentz force microscopy[15], scanning tunnelling microscopy[20] and soft X-ray magnetic circular dichroism (XMCD) adapted for microscopy[21–24]. All these techniques focus on the spin magnetism, ignoring the orbital aspect.

In this work, we consider three aspects of the chirality-driven orbital moment physics. First, we characterize their properties using symmetry and the underlying electronic structure, clearly separating the SOC-driven from the chirality-driven orbital moments, by performing first-principles calculations for magnetic trimers. Then, we show that the chirality-driven orbital moments inherit the topological properties of the magnetic structure that generates them, focusing on skyrmionic structures. Last, we exploit the distinct properties of the chirality- and SOC-driven orbital moments to propose a new experimental protocol, based on XMCD, that can directly establish whether a given sample has a topological magnetic structure.

## Results

**Chirality-driven orbital moments in magnetic trimers.** A nonvanishing scalar spin chirality requires at least three magnetic atoms, so as prototypes we take magnetic trimers formed by Cr, Mn, Fe and Co atoms. They are supported on the Cu(111) surface (see Fig. 1a for the atomic structure), a common choice of substrate with weak SOC. The electronic structure of a target magnetic state is found using constrained density functional theory (DFT) calculations[25–27] (see Methods and Supplementary Note 1). Three kinds of magnetic structures are considered: ferromagnetic (F), chiral right-handed (R) and chiral left-handed (L) (Fig. 1b–d). If the orbital moments depend on the chirality of the magnetic state ($C_{123} \neq 0$), the R and L structures should show dissimilar behaviour; the F structures serve as reference ($C_{123} = 0$).

First, we consider the Fe trimer. Varying the polar angle $\theta$ from 0 to 90° brings the spin moments from pointing normal to the surface to lie in the surface plane. The spin moment of Fe #1 follows the same angular path for all three kinds of magnetic structures, as emphasized in Fig. 1b–d. In Fig. 2a, we focus on the projection of its orbital moment on the $z$ axis. We find a nearly $\cos \theta$ dependence for the F angular sweep, as the orbital moment follows the direction of the spin moment very closely. In contrast, the two chiral structures show opposite deviations from the F curve, with their average leading to a $\cos \theta$-like trend. Figure 2b reveals that the difference between the R and L curves has a distinct angular dependence. Redoing the calculations without SOC, the orbital moment for the F structure vanishes, while the orbital moments for R and L are equal in magnitude and opposite in sign, reproducing the orbital moment difference computed with SOC. Our results demonstrate that the orbital moment comprises two parts: a SOC-driven part, dominated by the local atomic SOC; and a nonlocal chirality-driven part ($C_{123}$), determined by the entire magnetic structure of the trimer. If the magnetic structure is chiral, that is, $C_{123} \neq 0$, but SOC is absent, the orbital moments persist.

Figure 2c–f shows the two nonvanishing components of the chiral orbital moment of atom #1, for the four different trimers. The properties of the chiral orbital moments depend on the electronic structure of the trimers (Supplementary Figs 1 and 2; Supplementary Note 1), in particular on which orbitals are near the Fermi energy. $Fe_3$ and $Co_3$ have partially filled $d$-states, with orbital degeneracies that can be easily lifted by the SOC or by the chiral magnetic structure, leading to sizeable orbital moments. In contrast, $Cr_3$ and $Mn_3$ are close to half-filling, and so the orbital moments are one order of magnitude smaller. The orientation of the chiral orbital moment depends additionally on the local site symmetry (here $C_s$; the global symmetry of the trimer is $C_{3v}$). Counterintuitively, and in contrast to the SOC-driven orbital moment, the chiral orbital moment is independent of the absolute

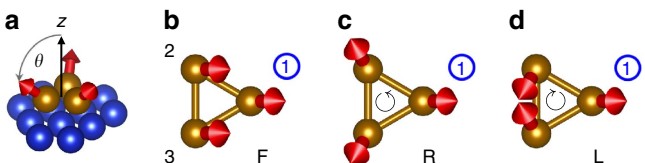

**Figure 1 | Atomic and magnetic structures of trimers on Cu(111).** (**a**) Atomic structure. The magnetic atoms are represented by golden spheres, and part of the Cu(111) substrate is represented by the blue spheres, indicating Cu atoms. The orientations of the spin moments are given by red arrows, and fixed as $\mathbf{n}_i = (\cos \phi_i \sin \theta, \sin \phi_i \sin \theta, \cos \theta)$ for $i = 1, 2, 3$. The surface normal defines the $z$ axis and the polar angle $\theta$. (**b–d**) Magnetic structures, top view ($\theta = 45°$). The choice of azimuthal angles $\phi_i$ defines the (**b**) ferromagnetic (F: $\phi_1 = \phi_2 = \phi_3 = 0°$), (**c**) chiral right-handed (R: $\phi_1 = 0°$, $\phi_2 = +120°$, $\phi_3 = -120°$) and (**d**) chiral left-handed (L: $\phi_1 = 0°$, $\phi_2 = -120°$, $\phi_3 = +120°$). Atom #1 (encircled) is chosen to have the same orientation of the spin moment in all structures.

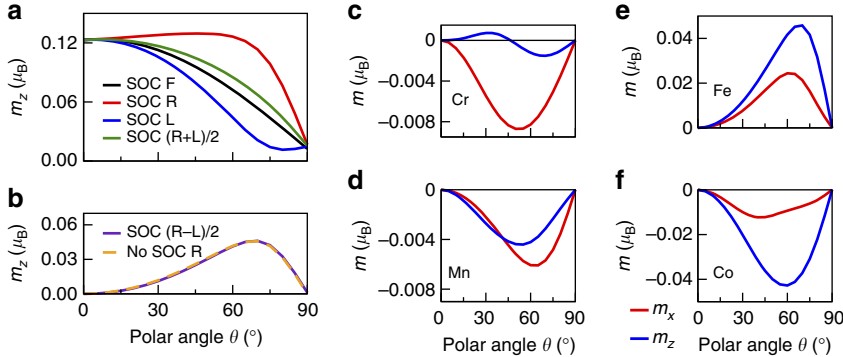

**Figure 2 | Orbital magnetic moment for atom #1 of trimers on Cu(111).** (**a**) Fe #1 in Fe trimer: variation of the $z$-component of the orbital moment, $m_z$, with the direction of the spin moment, for the ferromagnetic (F), chiral right- (R) and left-handed (L) magnetic structures (black, red and blue curves, respectively). The green curve shows the average of the data for the chiral structures. (**b**) Fe #1 in Fe trimer: difference between the orbital moments for the two chiral magnetic structures including spin–orbit coupling (SOC), and orbital moment for the R structure excluding SOC. (**c–f**) Nonvanishing components of the orbital moment for atom #1 in Cr, Mn, Fe and Co trimers, respectively, computed excluding the SOC, in the R structure.

real-space orientation of the spin moments: if all spin moments are rotated together in a way that leaves $C_{123}$ unchanged, the chiral orbital moment is also unchanged.

**Chirality-driven topological orbital moments in skyrmions.** Armed with our understanding of the chirality-driven orbital moments provided by the study of the trimers, we now investigate the role of the topology of the spin structure. Skyrmions are the most well-known topological magnetic structures, making them a natural target. After ref. 10, we define the spin structure by

$$\mathbf{n}(x, y) = (\cos(m\phi + \gamma)\sin\theta(r), \sin(m\phi + \gamma)\sin\theta(r), \cos\theta(r)),$$
(1)

assuming the skyrmion centre to be at the origin. In polar coordinates, we have $(x, y) = (r\cos\phi, r\sin\phi)$, $m$ is the vorticity, $\gamma$ is the helicity and $\theta(r)$ is the radial polar angle profile. An integer skyrmion number is obtained by integrating the emergent magnetic flux[10,11],

$$N_{sk} = \frac{1}{4\pi} \iint dx\, dy\, B_{sk}(x, y)$$
$$= \frac{1}{4\pi} \iint dx\, dy\, \mathbf{n}(x, y) \cdot \left(\frac{\partial \mathbf{n}}{\partial x} \times \frac{\partial \mathbf{n}}{\partial y}\right) = -m. \quad (2)$$

The rightmost integrand generalizes $C_{123}$ to the continuum case. For chiral skyrmions $N_{sk} = -1$, with $\theta = \pi$ in the centre. The skyrmions in MnSi are Bloch-like ($\gamma = \pi/2$) (ref. 10), while the skyrmions in Pd/Fe/Ir(111) are Néel-like ($\gamma = 0$) (ref. 20). As the skyrmion number is independent of the helicity, see equation 2, we set $\gamma = 0$ in the following. The skyrmion profile $\theta(r)$ has been experimentally determined for the Pd/Fe/Ir(111) system[20], and depends on the applied magnetic field (we provide a parametrization in Supplementary Note 2).

The interplay between the magnetic and electronic structure has attracted attention in connection to tunnelling experiments[28,29]. To analyse the orbital magnetism of skyrmions, we adopt mainly a tight-binding model, due to its transparency and ability to model large systems (see Methods and Supplementary Note 2). We consider a hexagonal unit cell, Fig. 3a, with periodic boundary conditions. The meaning of $N_{sk}$ in equation (2) is illustrated in Fig. 3b–e; its connection to the orbital moments will be examined in the following. For comparison, we also studied the largest skyrmions accessible with DFT[29,30]. These comprise 73 Fe atoms within an embedding cluster of 211 atoms, see Methods.

Figure 4a,b shows the emergent magnetic flux per site, $B_{sk}$ (equation (2)), which maps the local scalar spin chirality, for two skyrmion sizes. Figure 4c,d displays the corresponding orbital moment distributions, $m_{orb}$, computed directly by leaving out the SOC. The orbital moments are concentrated in regions of large noncollinearity, signalled by large $B_{sk}$. This confirms that the emergent magnetic field is the source of the chirality-driven orbital moments.

From Fig. 4c,d, the magnitude of the chirality-driven local orbital moments is seen to depend strongly on the skyrmion radius. However, the total chiral orbital moment does not, as shown in Fig. 5a, both for chiral skyrmions ($N_{sk} = -1$) and for the other skyrmionic structures sketched in Fig. 3b–e. Strikingly, the magnitude of the total chiral orbital moment is nearly constant, for $R_{sk} > 0.5$ nm. Figure 4 exposed the connection between $B_{sk}$ (equation (2)) and the orbital moments, so the total chiral orbital moment must inherit the topological properties of a skyrmionic structure, and thus be insensitive to deformations of the spin structure that preserve $N_{sk}$. Figure 5a also shows that the orbital moments are integer multiples of the total chiral orbital moment of the $N_{sk} = -1$ skyrmion. First-principles calculations for the largest attainable skyrmions (73 Fe atoms) confirm the existence of chiral orbital moments of the same order of magnitude as those found with the tight-binding model, see two data points in Fig. 5a.

**Experimental signatures of topological magnetic structures.** The first-principles calculations for the magnetic trimers and the tight-binding model for a skyrmion lattice show that the SOC-driven and the chiral contribution to the atomic orbital moments depend on the spin structure in distinct ways. Our calculations confirm that the SOC contribution to the atomic orbital moment is mostly driven by the local SOC of the corresponding atom. If the chiral contribution is absent (for example, the sample is ferromagnetic), there is a direct proportionality between the atomic orbital and spin moments, and thus between the net orbital and spin moments, $M_{orb} \propto M_{spin}$. This proportionality between the SOC-driven net orbital moment and the net spin moment should hold also for a noncollinear magnetic structure. An apparent deviation from this proportionality law is a signature of the presence of the chiral contribution to the orbital moment, and we propose that this can be exploited experimentally.

The XMCD sum rules[31–34] provide the net spin moment ($M_{spin}$) and orbital moment ($M_{orb}$) separately. These measurements are usually performed under an applied external

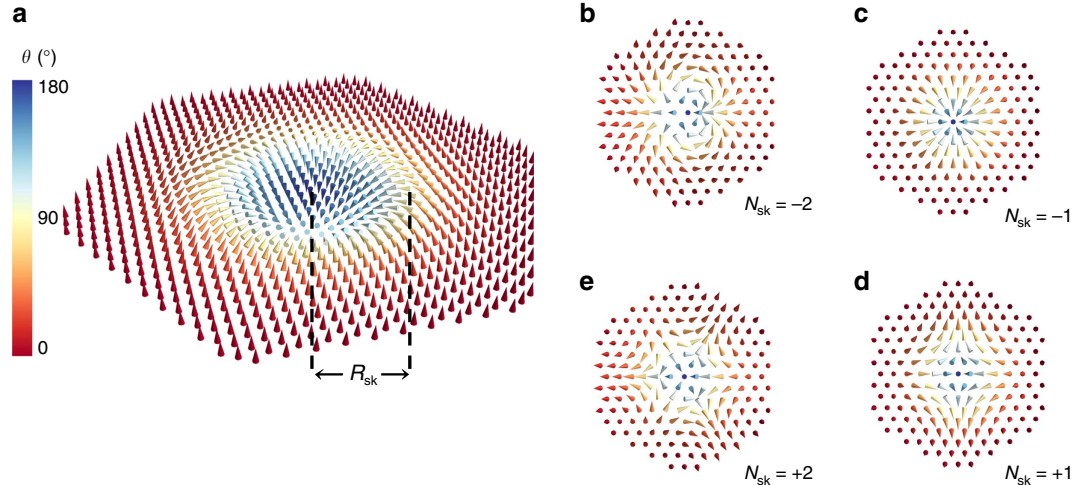

**Figure 3 | Skyrmionic magnetic structures.** (**a**) Hexagonal unit cell for the tight-binding model, containing 961 magnetic sites. It is 8.4 nm wide, using the lattice constant of the Pd/Fe/Ir(111) system. The cones show the orientation of the spin moments and are coloured by their polar angle. The skyrmion radius, $R_{sk}$, is defined by $\theta(R_{sk}) = 90°$. (**b-e**) Top view of structures with different skyrmion numbers, $N_{sk} = -2, -1, +1, +2$, respectively (equation 2; here $\gamma = 0$). The chiral skyrmions in Pd/Fe/Ir(111) correspond to $N_{sk} = -1$. Achiral skyrmions have $N_{sk} = +1$.

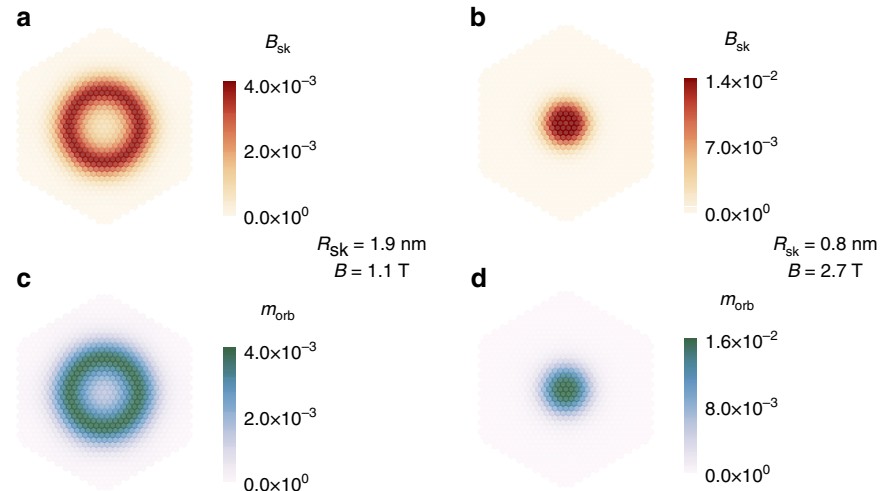

**Figure 4 | Chirality-driven orbital moments in skyrmions of different sizes ($N_{sk} = -1$).** (**a,b**) Emergent magnetic flux per site, $B_{sk}$ (equation (2)), for the skyrmion profiles characterized by $R_{sk} = 1.9$ nm and $R_{sk} = 0.8$ nm, corresponding to applied magnetic fields of $B = 1.1$ T and $B = 2.7$ T, respectively. (**c,d**) Orbital moment per site in $\mu_B$, $m_{orb}$, generated by the respective emergent magnetic fluxes shown in **a,b**, without SOC, computed using the tight-binding model for Pd/Fe/Ir(111).

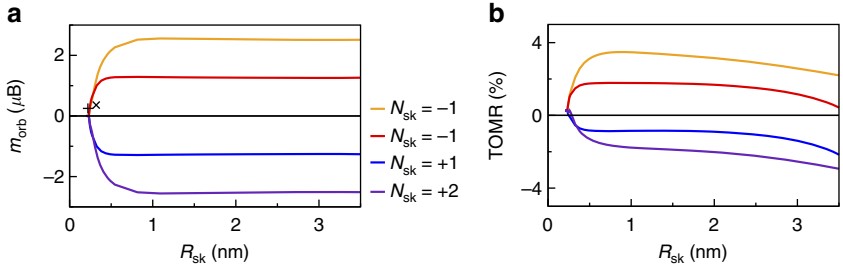

**Figure 5 | Signatures of chirality-driven orbital magnetic moments in skyrmions.** (**a**) Net chirality-driven orbital moments (computed without SOC) and (**b**) topological orbital magnetization ratio (TOMR; equation (3); computed with SOC), for the skyrmionic structures shown in Fig. 3b–e, using the tight-binding model with 961 magnetic sites. The skyrmion radius can be varied experimentally by changing the applied magnetic field. The TOMR can be detected in an XMCD experiment by comparing the skyrmion lattice phase with a reference ferromagnetic phase. Two additional data points in **a** mark the chirality-driven orbital moments from DFT calculations for the largest skyrmions first described in ref. 29, for Pd/Fe/Ir(111) ( × ) and Pd$_2$/Fe/Ir(111) ( + ), which contain 73 Fe atoms.

magnetic field, to ensure a net ferromagnetic component of the sample magnetization. The XMCD signal then leads to the average spin and orbital moments projected on the direction of incidence of the X-ray beam. All this applies equally well for a sample in a noncollinear magnetic state, as long as a net ferromagnetic component is present, which is ensured by the external field. The XMCD effect can also be used as a magnetic microscopy technique, as shown in refs 21–24.

We propose the following experimental protocol to detect the topological chiral orbital moments in a skyrmion-hosting sample, for instance, Pd/Fe/Ir(111). Applying a sufficiently large external magnetic field, the sample is driven to the field-polarized or ferromagnetic (F) state. XMCD measurements then provide the net spin and orbital moments, $M_{\mathrm{spin}}(\mathrm{F})$ and $M_{\mathrm{orb}}(\mathrm{F}) = \alpha M_{\mathrm{spin}}(\mathrm{F})$, where the orbital moment is driven purely by SOC. Reducing the strength of the applied magnetic field, the sample enters the skyrmion phase (Sk), and the net orbital moment is now $M_{\mathrm{orb}}(\mathrm{Sk}) = M_{\mathrm{orb}}(\mathrm{SOC}) + M_{\mathrm{orb}}(\mathrm{chiral}) \approx \alpha M_{\mathrm{spin}}(\mathrm{Sk}) + M_{\mathrm{orb}}(\mathrm{chiral})$. Here the main approximation is assuming the constant of proportionality $\alpha$ to be independent of the magnetic structure. The topological nature of the skyrmion spin structure generates the topological chiral contribution. We then expect a nonvanishing topological orbital magnetization ratio (TOMR) to be detected:

$$\mathrm{TOMR} = \frac{M_{\mathrm{orb}}(\mathrm{Sk})}{M_{\mathrm{orb}}(\mathrm{F})} - \frac{M_{\mathrm{spin}}(\mathrm{Sk})}{M_{\mathrm{spin}}(\mathrm{F})} \approx \frac{M_{\mathrm{orb}}(\mathrm{chiral})}{M_{\mathrm{orb}}(\mathrm{F})} \propto N_{\mathrm{sk}}. \quad (3)$$

An advantage of forming these ratios is that the unknown number of $d$-holes in the XMCD sum rules providing the net spin and orbital moments from the X-ray absorption intensities will mostly cancel out, assuming that they depend weakly on the magnetic state[31–34].

Figure 5b shows the expected behaviour of the TOMR using the tight-binding model of a skyrmion lattice with SOC, and comparison with Fig. 5a verifies the topological signature. Varying the applied magnetic field in the skyrmion phase (thus changing $R_{\mathrm{sk}}$) has a small impact on the TOMR, which further corroborates the topological origin. Thus, the detection of the TOMR requires no theoretical input, only the ability to drive a sample from a reference ferromagnetic state into a (possibly unknown) noncollinear state. If the TOMR is finite and also insensitive to changes in the noncollinear magnetic structure (for instance driven by the external magnetic field), it is a strong experimental sign of the topological character of the magnetic structure.

Comparing Fig. 5b with Fig. 5a also shows that the magnitude and sign of the TOMR determined by XMCD can be used to discriminate between chiral ($N_{\mathrm{sk}} = -1$) and achiral ($N_{\mathrm{sk}} = +1$) skyrmions, or more complex skyrmionic structures. As there is no simple rule for predicting even the sign of the TOMR, theoretical input on the properties of the electronic structure of the sample is needed to allow for definite conclusions on this point.

## Discussion

We have shown that orbital magnetic moments arise in magnetic materials not only from the spin–orbit interaction but also from the emergent magnetic field due to nonvanishing scalar spin chirality of the magnetic structure. The chirality-driven orbital magnetic moments have properties distinct from the SOC-driven ones, and have comparable magnitudes. The only requirement is a finite scalar spin chirality, so they should be present both in small clusters, wires, thin films and in bulk samples, as long as the spin structure is noncoplanar. For topological magnetic structures, the chirality-driven orbital magnetic moments inherit the underlying topology through the emergent magnetic flux that drives them, and so can be used to fingerprint skyrmionic

magnetic structures. These topological chiral orbital moments do not change appreciably under deformations of the spin structure that leave its topology unchanged, in contrast to the usual SOC-driven ones. They present a new way to characterize and investigate candidate materials for skyrmionic applications, via experimental determination of their orbital magnetic properties, with soft X-ray or other appropriate optical measurements. From a different perspective, comparing the spin and orbital magnetic moment distributions yields a real-space map of the emergent magnetic field in topological magnetic structures.

## Methods

**First-principles electronic structure calculations.** First-principles calculations for the trimers and skyrmions are performed within DFT, as implemented in the Korringa–Kohn–Rostoker Green function method[27,30], employing the scalar-relativistic approximation and the local spin density approximation parametrized by Vosko, Wilk and Nusair[35]. The SOC hamiltonian around each atom, $\mathcal{H}_{\mathrm{SOC}} = \xi \mathbf{L} \cdot \mathbf{S}$, is self-consistently included when required. We make use of a real-space embedding procedure, where the trimers or the isolated magnetic skyrmions are embedded in a non-perturbed host. For the trimer case, the host is the Cu(111) surface, while for skyrmions, two types of hosts are considered: $\mathrm{Pd}_n/\mathrm{Fe}/\mathrm{Ir}(111)$ ($n = 1,2$) in their ferromagnetic phase[29]. The skyrmion spin structure is self-consistently determined with SOC, for an embedded cluster containing 73 Fe atoms and 211 atoms in total. To extract the chiral orbital moments for the skyrmions, one single iteration is performed without the SOC. A brief discussion of the electronic structure calculations for trimers is given in Supplementary Note 1.

**Tight-binding calculations for large skyrmions.** The tight-binding model for the skyrmionic structures is constructed using the density of states from ferromagnetic DFT calculations for Pd/Fe/Ir(111). We consider two orbitals, $|x^2 - y^2\rangle$ and $|xy\rangle$, degenerate for the hexagonal lattice ($C_{3v}$ symmetry) and a Hamiltonian with local exchange coupling to prescribed spin directions and hopping to nearest neighbours only. We employ the skyrmion profile extracted in ref. 20. A brief discussion of the rationale behind the model and its construction is given in the Supplementary Note 2.

**Code availability.** The tight-binding code that supports the findings of this study is available from the corresponding authors on request.

**Data availability.** The data that support the findings of this study are available from the corresponding authors on request.

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

## Acknowledgements

M.d.S.D. would like to thank B. Dupé, S. Heinze and Y. Mokrousov for insightful discussions. This work is supported by the HGF-YIG Programme VH-NG-717 (Functional Nanoscale Structure and Probe Simulation Laboratory-Funsilab) and the ERC Consolidator grant DYNASORE. S.B. acknowledges funding from the European Union's Horizon 2020 research and innovation programme under grant agreement number 665095 (FET-Open project MAGicSky). The authors are grateful for the generous supercomputing resources provided by the Forschungszentrum Jülich.

## Author contributions

M.d.S.D. uncovered the chirality-driven orbital moments in DFT calculations for magnetic trimers, and developed and implemented the tight-binding model for skyrmions. J.B. and M.B. performed the DFT calculations for skyrmions and provided input for the construction of the tight-binding model. All authors discussed the results and helped writing the manuscript.

## Additional information

**Competing financial interests:** The authors declare no competing financial interests.

