## [Peer Review File · Nature Communications]

Reviewers' comments:

Reviewer #1 (Remarks to the Author):

Report on "Chirality-driven orbital magnetic moments: fingerprints of topological magnetic structures" by dos Santos Dias et. al

The manuscript in question presents a theoretical investigation of the emergence of chirality-driven orbital magnetic moments and a discussion about how this orbital magnetization may be used to characterize topological magnetic structures. Based on established DFT and tight-binding calculations, the presented results appears reliable and the conclusion that orbital magnetic moments can occur without the presence of spin-orbit coupling (SOI) is intriguing.

This particular result would be valuable to communicate further to researchers in the field since it fits well into the currently lively discussion about emerging magnetic fields and torques, and topological magnetism. The issue here is however the warranted novelty of the results since, as the authors honestly states, orbital moments without SOI has already been reported (see Refs. 10 and 14 in the manuscript). Granted, the current manuscript adds more details on the mechanism behind the emerging orbital moments, by analyzing the local density of states of the magnetic atoms, but the novelty can still be questioned.

The second part of the manuscript regards the possibility of using the chirality-induced orbital moments for characterizing topological magnetic structures, here skyrmions. The idea of being able to probe the skyrmion number experimentally is interesting and as suggested, the chirality-induced orbital moments might be used for this purpose. Unfortunately, as formulated in the manuscript it is not very clear how that "fingerprinting" would be performed in practice.

In particular I have the following questions/comments regarding the manuscript:

In the current work (and in Ref. 10), the systems where non-SOI orbital moments are observed have magnetic structures that seem to be determined by SOI effects like magnetocrystalline anisotropy (MAE) or Dzyaloshinskii-Moriya interactions (DMI). The considered trimers in Fig. 1 seem to only have a finite chiral contribution to the orbital moments for "conical structures" ($0 < \theta < 90$) and without MAE and DMI the ground state for the trimer would be either collinear for ferromagnetic exchange interaction or a planar Néel state for a antiferromagnetic exchange interaction. The "conical structures" are stabilized with constraining fields. The same could be said about the studied skyrmion structure, where the magnetic order seems to be set by a given radial profile (S3 in Supp. Note 2). Thus it would be nice to have a clarification about if the non-SOI orbital moments really can occur in ground-state configurations or only in excited configurations.

Can the emerging non-SOI orbital moments be seen as a surface effect or could it be found in bulk systems as well?

If the spin chirality causes non-SOI orbital moments, can it also cause other effects that are otherwise associated with SOI? Ref. 10 discusses anomalous Hall effects without SOI but what about other SOI effects?

For the proposed probing of skyrmion configurations, can it really be assumed that the SOI orbital moment follows ("tracks") the net spin moment?

It is not really clear how Fig. 5 could be used to fingerprint the skyrmion structures. Should the XMCD

measurements be performed for different applied fields and can the trend of Fig 5.b then be used as a template for the structure, or are detailed electronic structure calculations needed for every material/radius considered?

Thus, while the considered manuscript is of high quality, I can not recommend it for publication in Nature Communications in its present form. But as the premise of the study is of large interest it could be possible to improve the novelty of the work by further clarifying how the fingerprinting measurement could be done and by illustrating how the present findings about the non-SOI orbital moments differs from those reported in Refs. 10 and 14.

Reviewer #2 (Remarks to the Author):

This paper reports orbital moments driven by spin chirality in triangular spin trimers and full-size skyrmions. I find the idea interesting and its computational analysis neat. However, I have doubts about technical feasibility of the experiments proposed in this work, and about its implications in general.

1. The authors suggest that the topology of the orbital moments can be probed via XMCD measurements. To the best of my knowledge, the application of sum rules and the discrimination between spin and orbital moments in XMCD require that magnetic moments are polarized. Therefore, XMCD measurements used to determine spin and orbital moments are typically performed in the applied magnetic field. How should one understand the proposed XMCD measurement on the skyrmion phase, and which spin/orbital moments (or perhaps their projections on the field direction?) can be measured in this experiment?

A related question: what is the meaning of $M(\text{spin})$ and $M(\text{orb})$ in Eq. (3)? Are they full moments of a skyrmion, or projections of the moments?

2. Eq. (3) and Fig 5b are misleading, because spin and orbital moments are not directly proportional to any experimentally measurable intensity. In XMCD, spin and orbital moments are obtained from special combinations of the absorption coefficients for left- and right- polarized radiation measured on different absorption edges [see, for example, Eqs. (1) and (2) in PRL 75, 152 (1995)]. If the authors want to make direct link to the experiment, the realistic and experimentally relevant quantities, such as $I(+)$ and $I(-)$, have to be calculated.

3. I am not sure about general implications of these results. The discrimination between spin and orbital moments in XMCD is not an easy experiment, but what new information can one learn from such a measurement? What would be advantages of this approach compared to other relevant experimental techniques, such as SANS, Lorentz microscopy, STM, and topological Hall effect measurements?

More technical comments:

4. What exactly are calculations 'without the SOI'? Does it mean scalar-relativistic approximation?

5. The parametrization of the tight-binding model requires better justification. The few DFT data points

on Fig. 5 are hardly sufficient to verify the tight-binding model. Can one provide a more extensive and stringent test of this model and its parameters?

6. Strictly speaking, XMCD is not an optical technique because it operates at energies much higher than the standard optical (visible) spectral range.

7. As a side note, I found this manuscript very difficult to read because of excessive and sometimes exotic abbreviations. In my opinion, writing 'spin moment' and 'orbital moment' is much more natural than SMM and OMM, and there is enough space to write the 'spin-orbit coupling' in full, or at least use SOC as its more common abbreviated version.

REVIEWERS' COMMENTS:

Reviewer #1 (Remarks to the Author):

The authors have considered and addressed the earlier comments raised by both referees in thorough and clear way. As a result, the quality of the manuscript has now improved and from my point, the major doubts about publishing this manuscript in Nat. Comm. have been removed. Thus I can gladly recommend the revised manuscript for publication.

Reviewer #2 (Remarks to the Author):

I believe that the paper has been improved significantly, but there is one point in the XMCD part where a further clarification is needed. On page 9 after Eq. (3), I read ``An advantage of forming these ratios is that unknowns in the XMCD sum rules providing the net spin and orbital moments from the x-ray absorption intensities will mostly cancel out'. Please, clarify this statement. The standard simplification of the sum rules is the calculation of the ratio $m(\text{orb})/m(\text{spin})$ instead of evaluating $m(\text{orb})$ and $m(\text{spin})$ independently. But in your expression the ratios like $M_{\text{orb}}(S_k)/M_{\text{orb}}(F)$ are involved, so that both spin and orbital moments should be evaluated in each of the phases. What are the unknowns and how do they cancel out? This should be explained at least in the Supplemental Material.

We would like to thank both reviewers for the careful reading of our manuscript and for the many insightful comments and observations. In the following we present our detailed replies to all the points that have been raised.

Reviewer #1: The manuscript in question presents a theoretical investigation of the emergence of chirality-driven orbital magnetic moments and a discussion about how this orbital magnetization may be used to characterize topological magnetic structures. Based on established DFT and tight-binding calculations, the presented results appear reliable and the conclusion that orbital magnetic moments can occur without the presence of spin-orbit coupling (SOI) is intriguing.

This particular result would be valuable to communicate further to researchers in the field since it fits well into the currently lively discussion about emerging magnetic fields and torques, and topological magnetism. The issue here is however the warranted novelty of the results since, as the authors honestly states, orbital moments without SOI has already been reported (see Refs. 10 and 14 in the manuscript [now Refs. 6 and 7]). Granted, the current manuscript adds more details on the mechanism behind the emerging orbital moments, by analyzing the local density of states of the magnetic atoms, but the novelty can still be questioned.

The second part of the manuscript regards the possibility of using the chirality-induced orbital moments for characterizing topological magnetic structures, here skyrmions. The idea of being able to probe the skyrmion number experimentally is interesting and as suggested, the chirality-induced orbital moments might be used for this purpose. Unfortunately, as formulated in the manuscript it is not very clear how that "fingerprinting" would be performed in practice.

Reply: What we claim as novel is the topological character of the chirality-driven orbital magnetic moments, when the underlying magnetic structure itself has a non-trivial topology. This means that there is a contribution to the orbital moment that is constant under deformations of the magnetic structure, in contrast to the usual SOC contribution. The second part of the manuscript explores this, focusing on magnetic skyrmions. The proportionality between the net chiral orbital moment of a skyrmionic magnetic structure and the topological charge of the magnetic structure is presented, which we believe is a completely novel connection. Finally, we describe how the presence of the topological chiral orbital moments can be inferred from a XMCD experiment, which is also a novel experimental protocol. We have improved the discussion of the XMCD protocol, explaining what can be done without theoretical input or knowing the specifics of the magnetic structure under consideration. We thank the reviewer for pointing out the need for a better presentation of this part of our work.

Although chirality-driven orbital moments by themselves are not novel, as seen from Refs. 6 and 7, they are still a very unfamiliar concept to the vast majority of the scientific community, and they are poorly characterized in the existing literature. Thus we dedicated the first part of the manuscript to the task of unveiling the properties of these chiral orbital moments. There are also novel aspects in this part of the work, such as how to distinguish between the SOI-driven and the chirality-driven orbital moments, the detailed connection to the electronic structure, and the angular dependence of the chiral orbital moments. We have expanded the introductory discussion to highlight these aspects, in relation to Refs. 6 and 7.

Reviewer #1: In the current work (and in Ref. 7), the systems where non-SOI orbital moments are observed have magnetic structures that seem to be determined by SOI effects like magnetocrystalline anisotropy (MAE) or Dzyaloshinskii-Moriya interactions (DMI). The considered trimers in Fig. 1 seem to only have a finite chiral contribution to the orbital moments for "conical structures" ($0 < \theta < 90$) and without MAE and DMI the ground state for the trimer would be either collinear for ferromagnetic exchange interaction or a planar Néel state for a antiferromagnetic exchange interaction. The "conical structures" are stabilized with constraining fields. The same could be said about the studied skyrmion structure, where the magnetic order seems to be set by a given radial profile (S3 in Supp. Note 2).

Thus it would be nice to have a clarification about if the non-SOI orbital moments really can occur in ground-state configurations or only in excited configurations.

Reply: As we discuss in the manuscript, if the magnetic structure has finite local spin chirality, i.e. $S_1 \cdot (S_2 \times S_3)$ is finite, then the chiral orbital moments are present. Ground state magnetic structures where the magnetic moments do not all lie in the same plane can be stabilized by relativistic magnetic interactions, such as the MAE or the DMI, or by higher-order exchange interactions, such as the biquadratic or four-spin interactions, which are not of relativistic origin [see Ref. 5]. Another possibility is to engineer competing exchange interactions, for instance by creating a trimer with internal antiferromagnetic interactions but coupled ferromagnetically to a ferromagnetic substrate [see Ref. 4]. Non-coplanar magnetic ground state structures stabilized by the SOI will have two contributions to the orbital moments, the usual SOI-driven and the chirality

driven one. If SOI effects are negligible and the non-coplanar magnetic ground state is stabilized by biquadratic or four-spin interactions, the orbital moments are not zero and should be mostly chirality-driven. We have expanded the introduction to include this discussion.

Reviewer #1: Can the emerging non-SOI orbital moments be seen as a surface effect or could it be found in bulk systems as well?

Reply: The main requirement is a non-vanishing local spin chirality, i.e. $S_1 \cdot (S_2 \times S_3)$ not zero, as we emphasize in the manuscript. If a bulk system can host a non-coplanar ground state, for example under the conditions given in the previous answer, then local non-SOI orbital moments should be present. If the non-coplanar ground state has a net spin chirality, i.e. the sum of all $S_1 \cdot (S_2 \times S_3)$ is finite, then also a finite net non-SOI orbital moment should exist.

Reviewer #1: If the spin chirality causes non-SOI orbital moments, can it also cause other effects that are otherwise associated with SOI? Ref. 7 discusses anomalous Hall effects without SOI but what about other SOI effects?

Reply: We discuss in our work an orbital XMCD signal that is present also without SOI, although of course it follows from the non-SOI orbital moments. SOI effects that depend on the orientation of the spin moments with respect to the real space directions, such as the magnetic anisotropy energy or the anisotropic magnetoresistance, cannot be generated by the spin chirality. We are unsure of what other SOI effects the reviewer might have had in mind.

Reviewer #1: For the proposed probing of skyrmion configurations, can it really be assumed that the SOI orbital moment follows ("tracks") the net spin moment?

Reply: The local SOI orbital moment tracks the local spin moment, so the net SOI orbital moment also tracks the net spin moment. The DFT calculations for the trimers show that the misalignment between the SOI local orbital moment and the spin moment on the same atom is only of a few degrees at most. We expect this conclusion to extend to the case of magnetic skyrmions built out of magnetic $3d$ elements, as employed for our trimers.

Reviewer #1: It is not really clear how Fig. 5 could be used to fingerprint the skyrmion structures. Should the XMCD measurements be performed for different applied fields and can the trend of Fig 5.b then be used as a template for the structure, or are detailed electronic structure calculations needed for every material/radius considered?

Reply: The fingerprint is the excess intensity ratio between the orbital and the spin signals. Varying the applied magnetic field is useful to change the magnetic structure, for instance by modifying the skyrmion radius. If the excess intensity ratio is insensitive to changes in the magnetic structure, as we demonstrate by varying the skyrmion radius, then it must have a topological character. All this can be done experimentally without theoretical input. For the details of the magnitude and sign of the effect electronic structure information is needed. The trend shown in Fig. 5 has illustrative value; for other magnetic systems the actual dependence on the magnetic field or skyrmion radius might be different. We have expanded the discussion in the manuscript to clarify these points.

Reviewer #1: Thus, while the considered manuscript is of high quality, I can not recommend it for publication in Nature Communications in its present form. But as the premise of the study is of large interest it could be possible to improve the novelty of the work by further clarifying how the fingerprinting measurement could be done and by illustrating how the present findings about the non-SOI orbital moments differs from those reported in Refs. 6 and 7.

Reply: We believe we have allayed the concerns of the reviewer. We have expanded the discussion of the XMCD fingerprinting protocol, as mentioned in the previous answer. Concerning the overall novelty of our findings, we believe this is made clear by our first answer to the reviewer, and by our expanded introduction to the manuscript.

Reviewer #2: This paper reports orbital moments driven by spin chirality in triangular spin trimers and full-size skyrmions. I find the idea interesting and its computational analysis neat. However, I have doubts about technical feasibility of the experiments proposed in this work, and about its implications in general.

1. The authors suggest that the topology of the orbital moments can be probed via XMCD measurements. To the best of my knowledge, the application of sum rules and the discrimination between spin and orbital moments in XMCD require that magnetic moments are polarized. Therefore, XMCD measurements used to determine spin and orbital moments are typically performed in the applied magnetic field. How should one understand the proposed XMCD measurement on the skyrmion phase, and which spin/orbital moments (or perhaps their projections on the field direction?) can be measured in this experiment?

Reply: Our proposed experimental protocol requires the existence of net spin and orbital moments, so the XMCD signal is finite and can be used to measure them. This is not an essential restriction: for instance, a skyrmionic structure has a net spin moment. The application of an external magnetic field is not detrimental. On the contrary, varying the applied field is essential to deform the non-collinear magnetic structure of the sample, and to ascertain whether there is an excess of orbital magnetism insensitive to deformations of the magnetic structure. This would be the key signature of its topological origin. We have expanded the discussion of the XMCD fingerprinting protocol in the main text to clarify the rationale behind our proposal and address these questions.

Reviewer #2: A related question: what is the meaning of $M(\text{spin})$ and $M(\text{orb})$ in Eq. (3)? Are they full moments of a skyrmion, or projections of the moments?

Reply: They correspond to the net spin and orbital moments of the skyrmion, which are normal to the plane of the magnetic atoms. We have clarified this in the main text.

Reviewer #2: 2. Eq. (3) and Fig 5b are misleading, because spin and orbital moments are not directly proportional to any experimentally measurable intensity. In XMCD, spin and orbital moments are obtained from special combinations of the absorption coefficients for left- and right- polarized radiation measured on different absorption edges [see, for example, Eqs. (1) and (2) in PRL 75, 152 (1995)]. If the authors want to make direct link to the experiment, the realistic and experimentally relevant quantities, such as $I(+)$ and $I(-)$, have to be calculated.

Reply: For our fingerprinting protocol, the required input quantities are the net spin and orbital moments of the magnetic structure. These are provided by the well-established XMCD sum rules, see Refs. 30-32 in the main text; the reviewer mentions Ref. 32. Thus we need not compute the experimental intensities, only to invoke the sum rules and work with the quantities that they provide. We have reformulated the presentation of the fingerprinting protocol to clarify these points.

Reviewer #2: 3. I am not sure about general implications of these results. The discrimination between spin and orbital moments in XMCD is not an easy experiment, but what new information can one learn from such a measurement? What would be advantages of this approach compared to other relevant experimental techniques, such as SANS, Lorentz microscopy, STM, and topological Hall effect measurements?

Reply: As we explain in the main text, our XMCD fingerprinting protocol provides a direct experimental verification of the topological nature of a magnetic structure. Presently, the only experimental measurement giving direct access to this information is the topological Hall effect. The other experimental techniques infer whether a magnetic structure is topological indirectly, by attempting to map the full 3D magnetic structure. We believe none of the experimental techniques mentioned by the reviewer can be called “easy”, which in itself is not a very informative term without further elaboration. We also believe that having a complementary XMCD-based technique to the transport measurements of the topological Hall effect would be of great value for the field. We have expanded the introduction to highlight these aspects.

Reviewer #2: 4. What exactly are calculations 'without the SOI'? Does it mean scalar-relativistic approximation?

Reply: The calculations without SOI correspond to the scalar-relativistic approximation for the DFT case, and to excluding the SOI term for the tight-binding case. We have added this information to the Methods.

Reviewer #2: 5. The parametrization of the tight-binding model requires better justification. The few DFT data points on Fig. 5 are hardly sufficient to verify the tight-binding model. Can one provide a more extensive and stringent test of this model and its parameters?

Reply: Supplementary Note 2 contains the details about the tight-binding parametrization, and the comparison with the DFT calculations of the Pd/Fe/Ir(111) skyrmion host system. Fig. S3(a) shows that the model can reproduce the bandwidth for each spin projection, and gives a fair account of the density of states near the Fermi energy. This is as close as we can get to the DFT calculations with such a simplified model. The DFT calculations marked in Fig. 5 of the main text correspond to the largest feasible skyrmion calculations (73 Fe atoms; the whole embedded cluster contains 211 atoms) – note that the model contains 961 magnetic sites. They are meant to indicate that our tight-binding model produces orbital moments with the correct order of magnitude when comparison with DFT is possible, while also highlighting that the regime of topological orbital magnetism is beyond the capabilities of present state of the art DFT calculations. We emphasize that the aim of the tight-binding model is not to reproduce the details of the DFT electronic structure, but to combine the minimum required ingredients for the manifestation of topological orbital magnetism in realistic skyrmionic magnetic structures.

Reviewer #2: 6. Strictly speaking, XMCD is not an optical technique because it operates at energies much higher than the standard optical (visible) spectral range.

Reply: This is entirely correct. The sentence at fault is the last line of the abstract: “Furthermore, it provides a new route for the experimental detection and characterization of topological magnetic structures, by optical means.” This has been changed to “by soft x-ray spectroscopy”.

Reviewer #2: 7. As a side note, I found this manuscript very difficult to read because of excessive and sometimes exotic abbreviations. In my opinion, writing 'spin moment' and 'orbital moment' is much more natural than SMM and OMM, and there is enough space to write the 'spin-orbit coupling' in full, or at least use SOC as its more common abbreviated version.

Reply: We apologize for this. The present submission to Nature Communications is not constrained by the stringent word limit under which the manuscript was prepared, so we have followed the recommendation of the reviewer and replaced abbreviations by the full words.

Reviewer #1: The authors have considered and addressed the earlier comments raised by both referees in thorough and clear way. As a result, the quality of the manuscript has now improved and from my point, the major doubts about publishing this manuscript in Nat. Comm. have been removed. Thus I can gladly recommend the revised manuscript for publication.

Reply: We would like to thank the Reviewer for all his insightful comments and suggestions, and for recommending our work for publication.

Reviewer #2: I believe that the paper has been improved significantly, but there is one point in the XMCD part where a further clarification is needed. On page 9 after Eq. (3), I read “An advantage of forming these ratios is that unknowns in the XMCD sum rules providing the net spin and orbital moments from the x-ray absorption intensities will mostly cancel out”. Please, clarify this statement. The standard simplification of the sum rules is the calculation of the ratio $m(\text{orb})/m(\text{spin})$ instead of evaluating $m(\text{orb})$ and $m(\text{spin})$ independently. But in your expression the ratios like $M_{\text{orb}}(S_k)/M_{\text{orb}}(F)$ are involved, so that both spin and orbital moments should be evaluated in each of the phases. What are the unknowns and how do they cancel out? This should be explained at least in the Supplemental Material.

Reply: We completely agree that such a statement requires clarification. We have changed the sentence pointed out by the Reviewer in the main text to the following: “An advantage of forming these ratios is that the unknown number of d-holes in the XMCD sum rules providing the net spin and orbital moments from the x-ray absorption intensities will mostly cancel out, assuming they depend weakly on the magnetic state [31-34].” The rationale in forming these ratios using measurements in different magnetic phases is thus the same as for the more usual $M_{\text{orb}}/M_{\text{spin}}$ ratio. We would like to thank the Reviewer for his thorough reading of the manuscript and for his many requests for clarification and explanation, which have greatly improved the final form of the manuscript.